# Surface Structure Effects on H and O Adsorption on Gold, Nickel and Platinum Nanoparticles

**DOI:** 10.3390/ma18030631

**Published:** 2025-01-30

**Authors:** Nadezhda V. Dokhlikova, Andrey K. Gatin, Sergey Y. Sarvadii, Dinara Tastaibek, Vladislav G. Slutskii, Maxim V. Grishin

**Affiliations:** 1N.N. Semenov Federal Research Center for Chemical Physics of Russian Academy of Sciences (FRCCP RAS), Kosygina Street 4, 119991 Moscow, Russia; akgatin@yandex.ru (A.K.G.); sarvadiy15@mail.ru (S.Y.S.); slutsky@list.ru (V.G.S.); mvgrishin68@yandex.ru (M.V.G.); 2Institute of Cybernetics and Information Technology, Satbaev University (KazNRTU), Satbaeva Street 22A, Almaty 050013, Kazakhstan; baimukhambetovadinara@gmail.com

**Keywords:** quantum chemistry, DFT, modelling, nanoparticles, adsorption, surface, gold, platinum, nickel

## Abstract

Using quantum chemical modelling, in this work, we considered the structure effects determining the adsorption of H and O atoms on (111), (100), (110) and (211) surfaces of gold, nickel and platinum nanoparticles. Surface deformation enhanced the adatom bonding to active sites with a large coordination number on flat (111) and (100) surfaces, while no distinct tendency was observed on kinked (110) and (211) surfaces. The effect of the neighboring atoms depends on the coupling matrix element $V_{ad}^{2}$. For metals with a considerable matrix element, the adsorption energy decreases with the rise in coordination number, and vice versa.

## 1. Introduction

Metal and oxide nanoparticles have received great interest from the scientific community over the past 20–30 years due to their unique and promising physical and chemical properties [1]. High reactivity in a wide range of reactions and chemical synergism of their surfaces make nanoparticles a promising basis for catalysts, electronics, sensors and other functional materials [2,3]. Among the various nano-objects, we must mention nanoparticles of transition metals such as nickel and platinum. Catalysts based on these transition metals are widely used in the chemical industry due to their high activity [4]. Gold-based nanostructures are also worth noting, exhibiting high reactivity due to their size effect, while massive gold is chemically inert [5].

Any catalyst is a complex inhomogeneous system, and various processes may occur on its surface, such as reacting molecule adsorption, migration of intermediates, excitation and relaxation of active sites, etc. The measured or calculated macroscopic parameters of such systems depend on a huge number of interrelated phenomena. Therefore, the study of such systems is a very complicated task. In order to establish the fundamental reasons for chemical activity, it is necessary to investigate in detail the evolution of the electronic and atomic structure of nanoparticle surfaces. Such experiments require well-defined nanostructured coatings and precisely measured doses of gas adsorbate. All these requirements are met by methods of scanning tunneling microscopy and spectroscopy (STM/STS) that allow us to use single nanoparticles as model objects interacting with simple gas adsorbates. At the same time, numerical modelling is a reliable supplement to STM/STS experiments. It can simulate the changes in the electronic and atomic structure of the adsorbate complex and reveal the mechanism of elementary acts of surface reactions [6]. The size of the real nanoparticles studied in the experiment was about 3–5 nm. Such nanoparticles consist of at least several hundred atoms. Such huge objects are beyond the capabilities of DFT modeling. On the other hand, it is quite possible to use small clusters as models of nanoparticles, although due to the limited size of the system, it may incorrectly reflect the features of the electronic structure desired for calculation. In this case, modeling nanoparticles using surfaces is the most acceptable approach, since the main emphasis here is on the qualitative comparison of changes in the electronic structures of model nanosystems in the experiment and simulation. In this work, a set of different surfaces simulates the different domains that make up the inhomogeneous surface of nanoparticles [7].

The results obtained from the STS measurements allow us to estimate the change in the local conductivity of nanoparticles during gas adsorption, i.e., the electronic structure. Similarly, when studying nanoparticle models using DFT methods, the emphasis is on determining changes in the electronic structure of the models. Since the simulation is carried out on objects based on transition and noble metals, it is more convenient to use the d-band center model [8,9,10]. Numerous studies have shown its applicability for various adsorbates interacting with transition metal surfaces with a well-defined structure [11,12,13].

The interpretation of such studies may be ambiguous due to the deep interrelation of the electronic and atomic structures of nanoparticles. In order to compare correctly the adsorption properties of metal-based nano-objects, uniform model systems are necessary. DFT modeling is a reliable and proven tool for simulating trends and qualitative changes in calculated quantities, if the uniformity and methodology are maintained while constructing models and selecting parameters [14]. However, even in such a case, the adsorbate bonding with a surface always leads to surface deformation, which may differ for various metals. The aim of the present work is to estimate this effect for three metals with various electronic structures—nickel, platinum and gold [15,16,17,18,19]. The choice of nickel, platinum and gold is due to two reasons. Firstly, these metals are widely used as a basis for model functional nanomaterials and were studied by our team using STM/STS methods. Secondly, the selected series of metals allows us to observe the relative change in adsorption properties when moving along the periodic table of elements from top to bottom from nickel to platinum and from left to right from platinum to gold. The set of surfaces was chosen based on the variation in the surface curvature (flat/kinked) and the possibility of modeling the cell within the DFT framework. The supercells of high-index surfaces, for example (321), contain over a hundred atoms, so two surfaces containing various features were chosen as high-index surfaces: (110) with a break, that is, alternating chains of protruding and depressed atoms, and (211) with a step, that is, an edge formed by low-coordinated atoms. Smooth surfaces (111) and (100) were used to model homogeneous sets of active sites with different packing, i.e., dense (111) and more sparse (100). When choosing the adsorbate, we also took into account the results of experiments where hydrogen and oxygen acted as a simple model adsorbate. In the simulations, two different adsorbates also made it possible to compare the adsorption properties of surfaces with adatoms of different types. For flat and kinked surfaces of these metals, we simulated the following processes: (i) adsorption of H and O atoms on highly symmetric active sites; (ii) adsorption of H and O atoms on active sites with fixed and relaxed surface atoms. Thus, in this article, we analyzed the results obtained and determined the effect of surface deformation on the adsorption energy.

## 2. Calculation Details

The quantum-chemical ab initio calculations were carried out in Quantum Espresso [20] using generalized gradient approximation with the Perdew–Burke–Ernzerhof (PBE) functional [21] with parametrization for solids (PBEsol) [22] in Vanderbilt-constructed ultra-soft pseudopotentials [23]. The maximum energy limit was set as 400 eV. The Brillouin zone integration was carried out using the Monkhorst–Pack special points method [24] with a 0.01 eV smearing width of the Methfessel–Paxton scheme [25]. The energy convergence criterion of the self-consistent field was set as 10^−6^ eV. The van der Waals dispersion interactions were taken into account when applying the Grimme method (DFT-D3) [26]. The relaxation of the atoms in the slab was simulated according to the Broyden–Fletcher–Goldfarb–Shanno (BFGS) algorithm, with the convergence criterion for energy optimization of the atom geometry equal to 10^−5^ eV and the maximum strength equal to 10^−4^ eV/Å. The translation periods of the original slabs were optimized.

The selected models were a set of low-index flat (111) and (100) surfaces, differing in symmetries, and high-index jagged (110) and stepped (211) surfaces, imitating the inhomogeneous structure of the real nanoparticle surface. We used (111), (100), (110) and (211) slabs consisting of 3 atomic layers with vacuum spacing of the order of 20 Å under periodic conditions to simulate the surface of Au, Ni and Pt nanoparticles. Models of (111) and (100) surfaces consisted of 48 metal atoms in the supercell: the model for the (110) surface consisted of 64 atoms, and the model for the (211) surface consisted of 54 atoms. The facet areas were of the order of 10 × 10 Å, depending on metal and orientation. These facet areas were able to sufficiently simulate single-atom adsorption since one atom disturbs the surface structure locally.

The simulation of adsorption properties of the above surfaces included the bonding energy calculation for H and O adatoms on highly symmetric active sites [9]:(1)$$E_{bond}^{f}\left(CN\right)=E_{ads}^{f}\left(CN\right)-E_{surf}-E_{a}$$(2)$$E_{bond}^{r}\left(CN\right)=E_{ads}^{r}\left(CN\right)-E_{surf}-E_{a}$$

Here, $E_{a}$ is the H or O adatom total energy, $E_{surf}$ is the slab total energy, and $E_{ads}\left(CN\right)$ is the total energy of the slab with adsorbate, depending on the coordination number *CN*. The bonding energy $E_{bond}\left(CN\right)$ was calculated both for slabs with fixed and relaxed metal atoms, being denoted by the upper index *f* and *r* correspondingly.

To investigate the diffusion possibility, the intermediary active sites were simulated. This allowed us to plot the energy diagram of H and O adatom diffusion over these surfaces. Calculated for fixed surface atoms, the adsorption energies were used to map the active sites of metal surfaces.

The bond length was defined as an equilibrium distance between the adatom and one-centered active sites. The bonding energy $E_{bond}^{f}\left(CN=1\right)$ allows to estimate the electronic effect on the adsorbate–surface interaction. Having calculated the bonding energy for multi-fold active sites with fixed slab atoms, one can estimate the effect of the neighboring atoms of the surface using the following equation [9]:(3)$$E_{nhb}=E_{bond}^{f}\left(CN=2,3,4\right)-E_{bond}^{f}\left(CN=1\right)$$

Optimizing the surface atom position and calculating the bonding energy, we can determine the surface deformation effect as follows:(4)$$E_{def}=E_{bond}^{r}\left(CN\right)-E_{bond}^{f}\left(CN\right)$$

## 3. Results

### 3.1. Electronic and Atomic Structure of the Surface

Bonding energies of H and O atoms at the T position—at the one-centered active site—are shown in Figure 1. One can see how the position of the *d*-band center of the nearest slab atom affects the bonding energy. If we compare the surfaces of different metals with the same packing (see Figure 1), we can notice the following: the closer the *d*-band center to the Fermi level, the stronger the bonding. This is consistent with the *d*-band center model. For example, the strongest bonding of an O atom with the surface (111) is observed for Ni with the *d*-band center at −1.5 eV. Gold with the *d*-band center at −3.3 eV demonstrates the weakest bonding with O atoms. According to the data [14], the *d*-band centers on bulk Ni, Pt and Au are at −1.29 eV, −2.25 eV and −3.56 eV, correspondingly.

At the same time, for surfaces with different packing of the same metal, no obvious correlation is observed (see Figure 1). This is due to the different symmetries of the active sites, i.e., different effects of neighboring atoms on the surfaces of different packing. In addition, some atoms of the kinked (110) and (211) surfaces are unequal in terms of their neighboring. For example, the adsorption sites of the (211) surface can be of various structures, since the adsorbent atom may have a various number of neighbors due to its location, i.e., at the edge, in the middle and in the corner of the atomic step.

Thus, the electronic effect, which is due to the *d*-band occupation and various positions of its center, will significantly influence the adatom bonding with the surface. However, the atomic effect must not be neglected, especially in the case of deformed and kinked surfaces of the same metal. The atomic effect here results from the surface deformation and neighboring atoms affecting the active site. The neighboring atoms influence is estimated by the bonding energies for active sites with various coordination numbers, when the positions of the metal atoms on the surface stay fixed. The optimization of the surface atoms’ position for the same active sites takes into account the effect of surface deformation caused by adsorption.

The flat (111) and (100) surfaces, in addition to the jagged (110) and the stepped (211) surfaces, have a different set of highly symmetric active sites (see Figure 2). At the (111) surface, there are one-fold T, two-fold B, and three- fold HCP and FCC sites (see Figure 2a). At the (100) surface, one can find one-fold T, two-fold B and four-fold H active sites (see Figure 2b). The stepped and jugged surfaces have a more varied structure. At the (110) surface, we have one-fold T1 and T2, two-fold Bl1, Bl2 and Bh1, Bh2 differing in bond lengths, and three-fold H1 and H2 sites (see Figure 2c). At the surface (211), there are even more types of active sites due to the step: one-fold T1, T2, T3, two-fold Bl1, Bl2, Bl3, Bl4, and Bh1, Bh2, Bh3 and Br, three-fold H1, H2, H3, H4, Hr, and four-fold R (see Figure 2d).

### 3.2. Highly Symmetric Sites of Nickel Surfaces

The results of adsorption and diffusion simulation for H and O atoms on nickel (111), (100), (110), and (211) surfaces are shown in Figure 3. There is a significant difference in the adsorption energies of H and O atoms, which is of the order of 2–4 eV (see Table 1).

One can conclude that bonding with any nickel surface is stronger for O adatoms than for H. This result corresponds to the STM/STS experiments, indicating that nickel nanoparticles remain oxidized even after a prolonged reduction in the sample in hydrogen at 600 K [27]. It can also be noticed that O bonding with the surfaces is more stable compared to H, since the diffusion barriers for O are higher. Potentially, H has higher mobility on nickel surfaces. Thus, nickel can be used as an accumulator and conductor of H. This is consistent with earlier works, where nickel was considered as a component of bimetallic nanostructures [28]. From the perspective of H adsorption and diffusion over nickel surfaces, O adatoms are some kind of catalytic poison, strongly bonding with surfaces and blocking the active sites.

In Figure 4, we can observe how the adsorption of H and O affects the projected density of states (PDOSs) of the nickel (111) surface atom. Here, the strong bonding of O is also indicated by the significant shift in the *d*-state density. Here, nickel demonstrates the highest reactivity as its coupling matrix element $V_{ad}^{2}$ is of minimal value among the platinum group [9]. For nickel, this results in negligibly weak repulsion and strong bonding even in the case of adsorbates with extended 2*p*-O orbitals.

The effect of the surface deformation on the adsorbate–surface bonding energy demonstrates different trends for flat (111) and (100) and kinked (110) and (211) nickel surfaces. As one can see in Figure 3a–d, for the H, HCP and FCC active sites, the bonding energy depends on the neighboring atoms and surface deformation more significantly than for the B and T active sites. That is, for flat (111) and (100) nickel surfaces, the atomic effect rises with the increase in the active site coordination number. Such a trend is quite expected, since the maximum distance is optimal between nickel surface atoms and the adsorbate, and this is what happens on multi-centered sites. This fact indicates a very small overlap between the adsorbate and adsorbent orbitals. Consequently, the repulsive contribution to bonding is also very small.

For the kinked (110) and (211) nickel surfaces, there is no such distinct correlation between the coordination number and the neighboring atom effect. At the same time, the effect of the surface deformation here is more significant, compared to the flat (100) and (111) surfaces—it is of an order of a few electron-volts. The first one is due to the charge distribution, which is less homogeneous on the kinked surface, changing the equilibrium distance between the adsorbate and the surface. The second one results from the fact that nickel atoms have a reduced set of neighbors on the kinked surface, and so their mobility is more vigorous. Thus, here a synergy takes place between the electronic and atomic effects. When interacting with adsorbates, kinked surfaces change their structure efficiently, due to the features of charge distribution, strengthening the adsorbate–surface bonding. Moreover, this effect is not local and manifests itself for all active sites.

### 3.3. Highly Symmetric Sites of Platinum Surfaces

The results of adsorption and diffusion simulation for H and O atoms on platinum (111), (100), (110), and (211) surfaces are shown in Figure 5. The difference between adsorption energies of H and O is of about 2 eV both for flat and kinked surfaces. We can conclude that O bonding with platinum surfaces is weaker than with nickel ones.

Both nickel and platinum belong to the same group of transition metals, having a similar electronic structure, but platinum has a considerable coupling matrix element $V_{ad}^{2}$. This results in an increase in Pauli repulsion when bonding, especially in the case of the adsorbates with extended orbitals like 2*p*-O [6]. For the adsorbate with a small orbital extension, like 1*s*, the repulsion effect is insignificant, and so platinum and nickel have similar bonding energies for H (see Table 2). The weak bonding between O and platinum is consistent with the results of STM/STS experiments. The reduction of oxidized platinum nanoparticles by hydrogen occurs even at room temperature, in contrast to the oxidized nickel nanoparticles [29]. The barrier against O migration on platinum is also lower by about 1 eV, compared to nickel. In Figure 6, we can observe how adsorption of H and O affects the PDOS of the Pt(111) surface atom. The large shift in *d*-states’ density upon adsorption of O correlates with strong bonding. The distribution of the *d*-states’ density for platinum is wider than for nickel. This indicates a significant $V_{ad}^{2}$ [9], since the interaction matrix element is proportional to the band width. One can distinguish various trends of atomic effects on bonding energy for flat (111) and (100) and kinked (110) and (211) platinum surfaces. For the smooth (111) and (100) surfaces, the effect of surface deformation correlates with the coordination number of the active site, as for the nickel surface. However, the effect of the neighboring atoms rises with the decrease in the coordination number. In the absence of surface deformation (see Figure 5a–d), one-fold T and two-fold B sites are more stable and differ insignificantly in bonding energy, while the bonding energy is much lower for the three-fold sites. The adsorbate tends to be located very close to the active site atoms, since the increase in the coordination number leads to strong repulsion. This is what makes the T active site more preferable despite the short bond length. We should also note that that surface deformation caused by adsorption can reverse this trend, since the surface atoms may change their position and the surface–adatom distance.

All the active sites of (110) and (211) surfaces are affected strongly by the surface deformation, and this is especially significant for the H1 site of the Pt(110). It should also be mentioned that for active sites with a small coordination number, there is a tendency to increase the bonding energy, although for smooth surfaces, it is more distinct. We can conclude that synergy of electronic and atomic effects also takes place for platinum surfaces, but the atomic effect on (110) and (211) surfaces is very weak. Its contribution varies in the range 0.5–1.0 eV for all kinked surfaces equally.

### 3.4. Highly Symmetric Sites of Gold Surfaces

The results of adsorption and diffusion simulation for H and O atoms on gold (111), (100), (110), and (211) surfaces are shown in Figure 7. Adatom bonding energies are the lowest for all gold surfaces, and this is consistent with the well-known fact of gold chemical inertness (see Table 3). According to the calculations, the dissociation energy is 4.53 eV for oxygen molecules and 3.31 eV for hydrogen molecules. These values exceed the energy of adatom bonding to flat (111) and (100) surfaces, and so the dissociative adsorption of hydrogen and oxygen is impossible [30]. Bonding is strong enough for some active sites of the kinked (110) and (211) surfaces, making them appropriate for dissociative adsorption. We should note that it is the kinked surfaces for which the surface deformation effect becomes crucial—it can drastically change the behavior of hydrogen and oxygen molecules. The surface deformation effect is especially distinct for the (110) surface. As one can see, disregarding the atomic effect, we obtain no active sites appropriate for dissociative adsorption. However, the bonding energies increase significantly in simulations that consider the surface deformation. For the stepped (211) surface, the atomic effect is less distinct, since even without it, there are some active sites with bonding energies sufficient for molecule dissociation. Nevertheless, all active sites become chemically active to adsorb hydrogen and oxygen, when surface deformation is taken into account. This is exactly what was observed in STM/STS experiments on hydrogen and oxygen adsorption on gold nanoparticles with inhomogeneous structures [31].

In Figure 8, we can observe how adsorption of H and O affects the PDOS of the Au(111) surface atom. Occupied outer *d*-shells and considerable orbital overlap matrix elements are known to be the reason for gold chemical inertness. The *d*-band of gold is evidently wider than the one of platinum and nickel, and its upper edge is approximately at −1 eV below the Fermi level. The adatom bonding to the *d*-band leads to the formation of two occupied states, with the anti-bonding state located at about −0.5 eV below the Fermi level. This confirms that the inertness of gold is the greatest compared with nickel and platinum. In the case of gold, the synergy of atomic and electronic effects can change the surface reactivity drastically.

For the densely packed (111) surface, which is the smoothest one, the effect of neighboring atoms is very small. The difference in adsorption energy for various active sites of the surface is about 0.1 eV (see Figure 7a,b). For the (100) surface, the effect of neighboring atoms increases for small coordination numbers. Here, the repulsive interaction prevents multi-centered bonding, as well as for platinum (111) and (100) surfaces. In this case, the deformation effect is significant, since it makes the multi-fold bonding more stable. For the rough (110) and (211) surfaces, the atomic effect does not depend distinctly on the coordination numbers of the active sites, due to the high mobility of atoms and inhomogeneous charge distribution.

## 4. Conclusions

In this paper, we investigated the atomic effects—surface deformation and neighboring atoms—determining the adsorption of H and O atoms on (111), (100), (110) and (211) surfaces of gold, nickel and platinum. We can conclude the following:For flat (111) and (100) surfaces, only the active sites with a large coordination number are affected significantly by surface deformation. For kinked (110) and (211) surfaces, the surface deformation effect is much stronger than for flat surfaces, though it has no drastic tendency. The surface deformation always results in strong bonding.The effect of neighboring atoms depends on the metal type and correlates indirectly with the coupling matrix element. The less the matrix element, the weaker the repulsion in metal–adatom complexes. In the case of weak repulsion, the effect of neighboring atoms increases with the coordination number of the active site. Strong repulsive interaction inverts this tendency.

## Figures and Tables

**Figure 1 materials-18-00631-f001:**
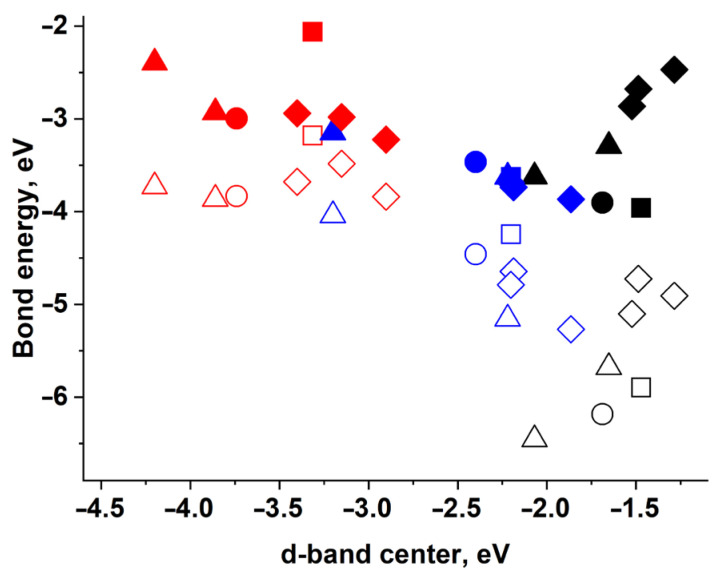
Bonding energy of H (solid figures) and O (empty figures) adatoms depending on the *d*-band center of the slab atoms: Ni (black figures); Pt (blue figures); Au (red figures). Various surfaces are represented: (111) surface (squares), (100) surface (circles), (110) surface (triangles) and (211) surface (rhombuses).

**Figure 2 materials-18-00631-f002:**
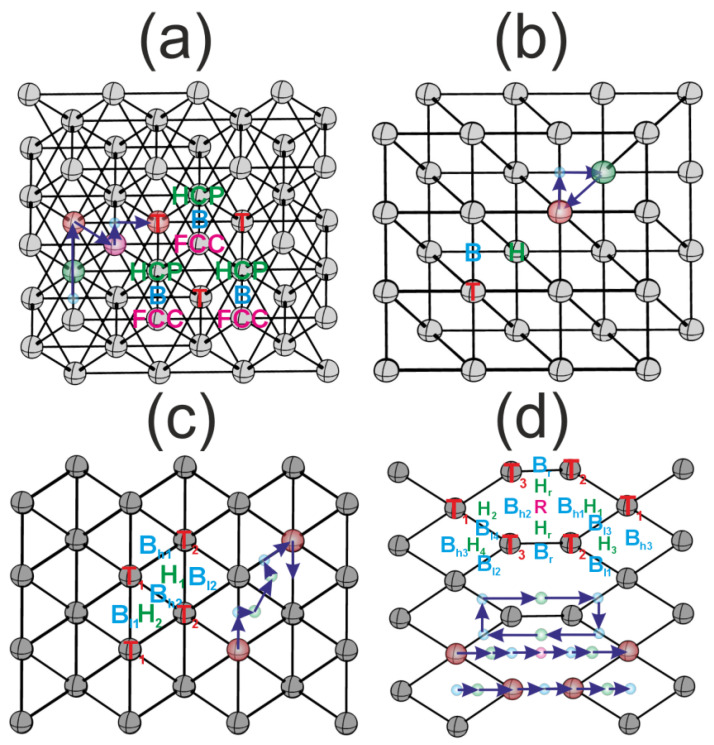
Highly symmetric active sites of various surfaces: (**a**) surface (111); (**b**) surface (100); (**c**) surface (110); (**d**) surface (211). The colors indicate active sites, T—red, B—blue, H, HCP—green, FCC—pink. The blue arrows show the diffusion paths.

**Figure 3 materials-18-00631-f003:**
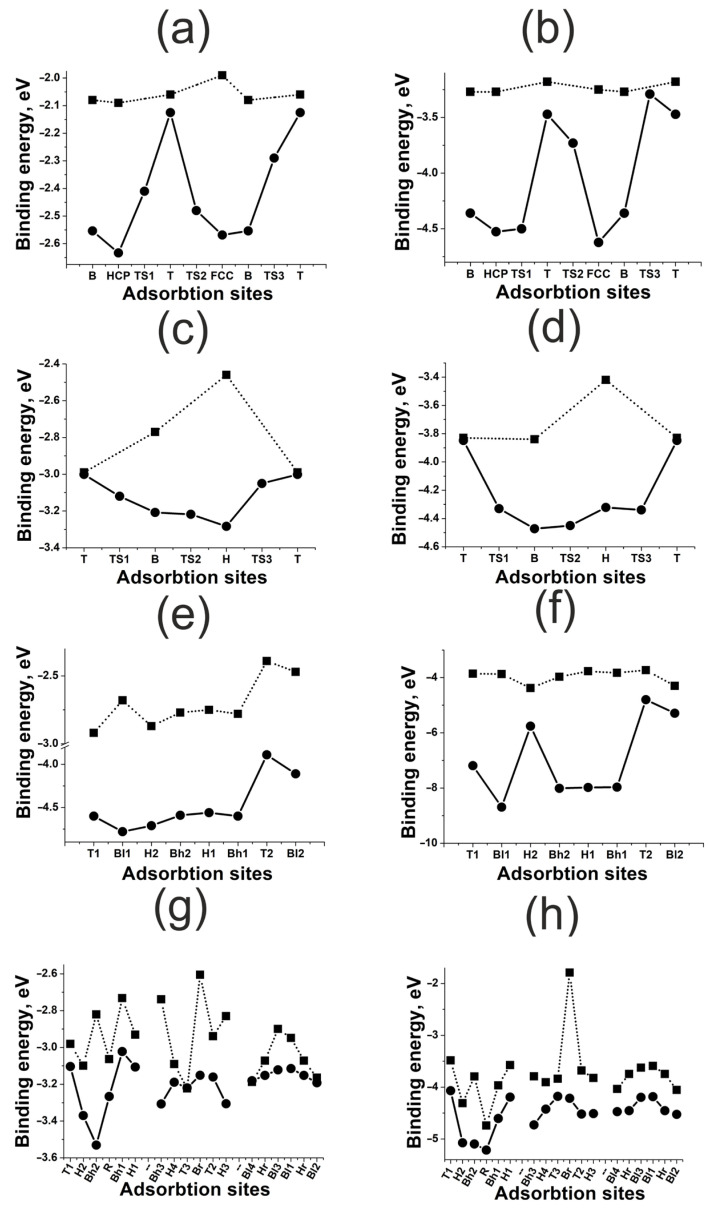
Energies of adsorption on active sites and diffusion energy diagrams for H and O atoms on nickel surfaces: (**a**) H on Ni(111); (**b**) O on Ni(111), (**c**) H on Ni(100); (**d**) O on Ni(100); (**e**) H on Ni(110); (**f**) O on Ni(110); (**g**) H on Ni(211); (**h**) O on Ni(211). The squares mark $E_{bond}^{f}$, the circles mark $E_{bond}^{r}$.

**Figure 4 materials-18-00631-f004:**
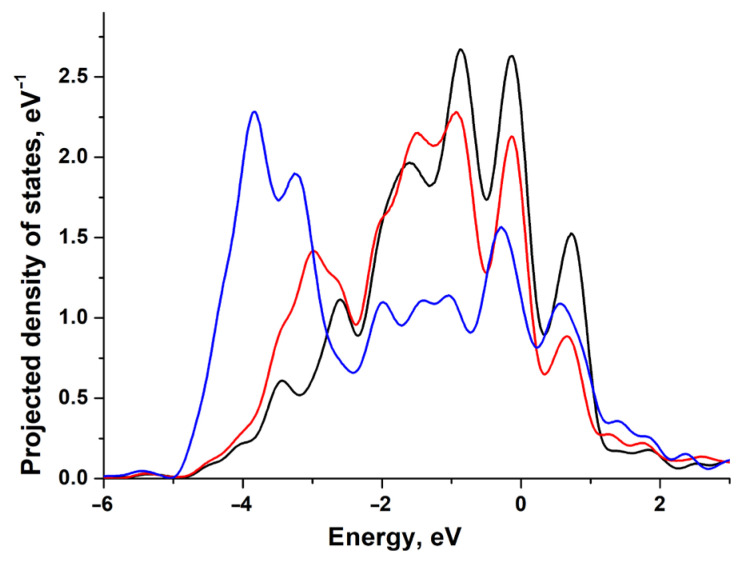
Projected density of states (PDOS) of the Ni(111) surface atom before adsorption (black line), after H adsorption (red line), and after O adsorption (blue line).

**Figure 5 materials-18-00631-f005:**
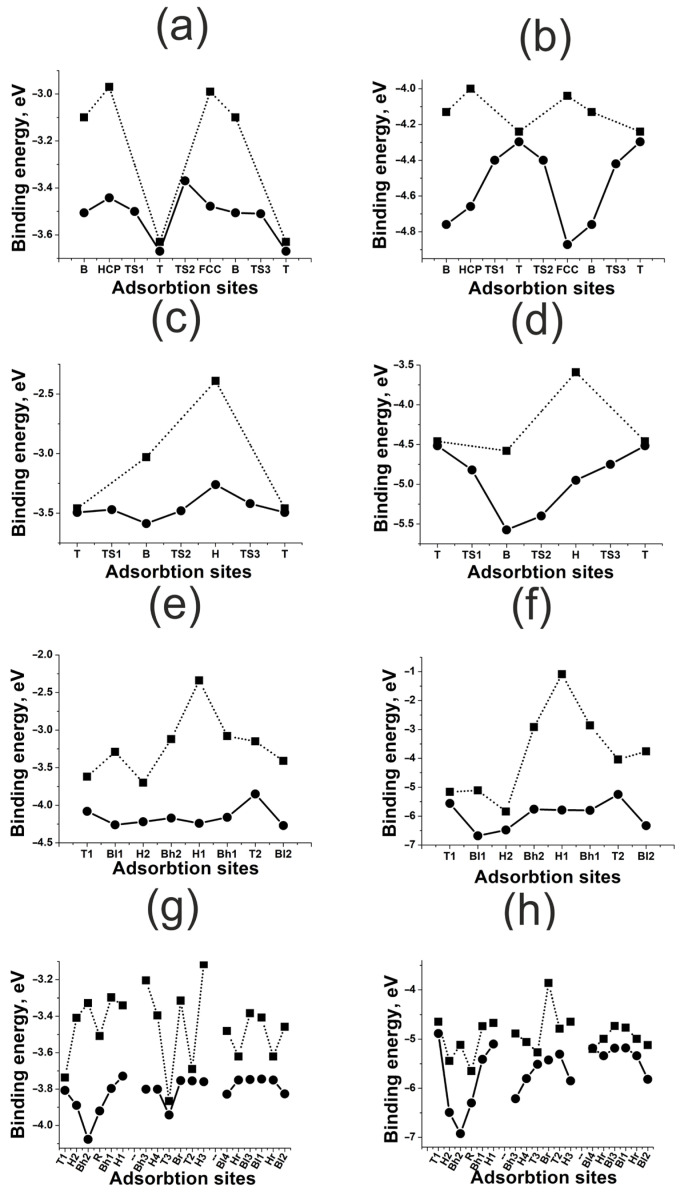
Energies of adsorption on active sites and diffusion energy diagrams for H and O atoms on platinum surfaces: (**a**) H on Pt(111); (**b**) O on Pt(111); (**c**) H on Pt(100); (**d**) O on Pt(100); (**e**) H on Pt(110); (**f**) O on Pt(110); (**g**) H on Pt(211); (**h**) O on Pt(211). The squares mark $E_{bond}^{f}$, the circles mark $E_{bond}^{r}$.

**Figure 6 materials-18-00631-f006:**
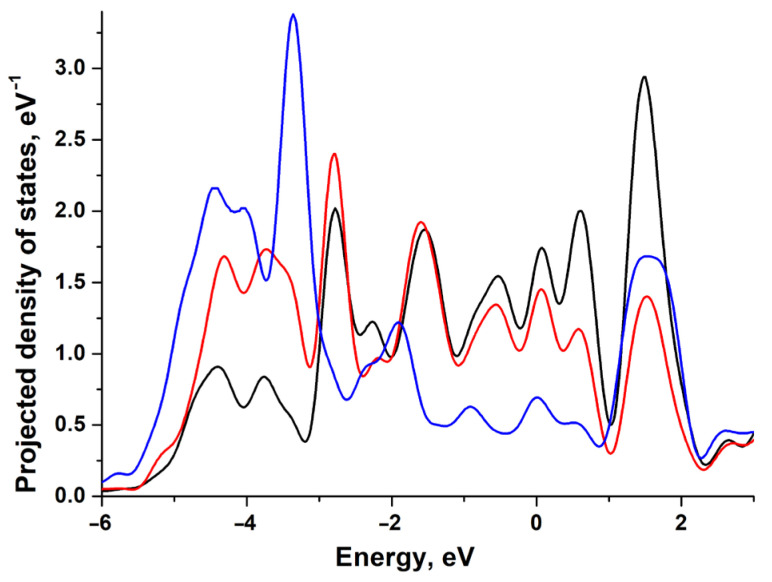
Projected density of states (PDOS) of the Pt(111) surface atom before adsorption (black line), after H adsorption (red line), and after O adsorption (blue line).

**Figure 7 materials-18-00631-f007:**
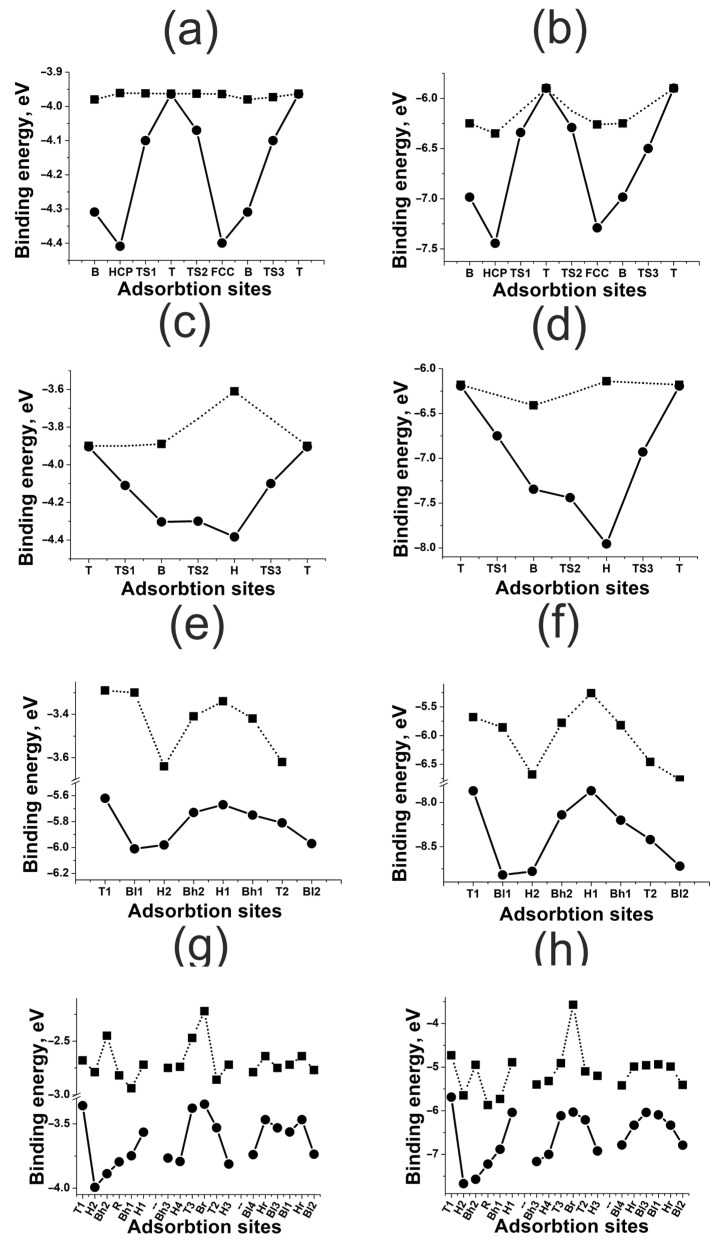
Energies of adsorption on active sites and diffusion energy diagrams for H and O atoms on gold surfaces: (**a**) H on Au(111); (**b**) O on Au(111); (**c**) H on Au(100); (**d**) O on Au(100); (**e**) H on Au(110); (**f**) O on Au(110); (**g**) H on Au(211); (**h**) O on Au(211). The squares mark $E_{bond}^{f}$, the circles mark $E_{bond}^{r}$.

**Figure 8 materials-18-00631-f008:**
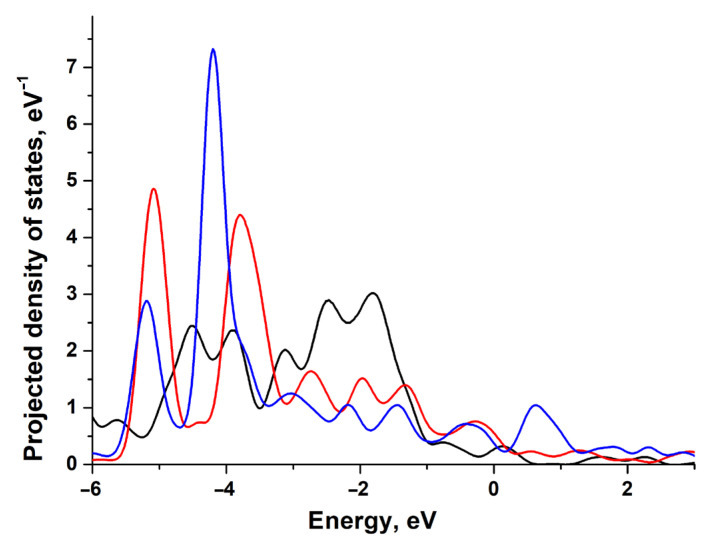
Projected density of states (PDOS) of the Au(111) surface atom before adsorption (black line), after H adsorption (red line) and after O adsorption (blue line).

**Table 1 materials-18-00631-t001:** Bonding energies $E_{bond}^{r}$(eV), bonding distances $R_{bond}^{r}$ (Å) and distances to the surfaces $R_{surf}^{r}$ (Å) of H and O adatoms for various active sites of Ni(111), Ni(100), Ni(110) and Ni(211) surfaces.

	H			O		
Ni(111)	$E_{bond}^{r}$	$R_{bond}^{r}$	$R_{surf}^{r}$	$E_{bond}^{r}$	$R_{bond}^{r}$	$R_{surf}^{r}$
T	–3.96	1.47	1.47	–5.90	1.67	1.67
B	–4.31	1.64	1.10	–6.98	1.797	1.32
FCC	–4.40	1.73	1.00	–7.29	1.86	1.22
HCP	–4.41	1.72	0.998	–7.44	1.86	1.21
Ni(100)	$E_{bond}^{r}$	$R_{bond}^{r}$	$R_{surf}^{r}$	$E_{bond}^{r}$	$R_{bond}^{r}$	$R_{surf}^{r}$
T	–3.90	1.46	1.46	–6.19	1.65	1.65
B	–4.30	1.61	1.07	–7.35	1.76	1.26
H	–4.38	1.84	0.665	–7.96	1.94	0.874
Ni(110)	$E_{bond}^{r}$	$R_{bond}^{r}$	$R_{surf}^{r}$	$E_{bond}^{r}$	$R_{bond}^{r}$	$R_{surf}^{r}$
T1	–5.62	1.48	1.48	–7.89	1.64	1.64
Bl1	–6.01	1.63	1.12	–8.82	1.76	1.27
H1	–5.67	1.95	1.44	–7.87	2.17	1.69
H2	–5.98	1.80	1.22	–8.78	1.97	1.47
Bh1	–5.75	1.77	1.34	–8.21	1.96	1.37
Bh2	–5.73	1.77	1.38	–8.14	1.95	1.44
Bl2	–5.98	1.71	1.21	–8.72	1.93	1.50
T2	–5.81	1.499	1.499	–8.42	1.73	1.73
Ni(211)	$E_{bond}^{r}$	$R_{bond}^{r}$	$R_{surf}^{r}$	$E_{bond}^{r}$	$R_{bond}^{r}$	$R_{surf}^{r}$
Bh1	–3.75	1.65	1.07	–6.88	1.85	1.35
H1	–3.56	1.79	1.12	–6.04	1.92	1.28
T1	–3.36	1.49	1.49	–5.69	1.69	1.69
H2	–3.99	1.71	1.02	–7.67	1.83	1.26
Bh2	–3.89	1.61	1.01	–7.57	1.73	1.19
R	–3.80	2.02	1.04	–7.22	2.25	1.46
Bl1	–3.56	1.63	1.14	–6.09	1.80	1.39
Bl2	–3.74	1.68	1.195	–6.79	1.798	1.39
Hr	–3.47	1.98	1.48	–6.33	2.19	1.71
T2	–3.53	1.53	1.53	–6.21	1.77	1.77
H3	–3.81	1.68	0.93	–6.92	1.84	1.26
Bh3	–3.77	1.62	1.02	–7.16	1.77	1.25
H4	–3.79	1.75	1.07	–7.00	1.85	1.22
T3	–3.38	1.49	1.49	–6.11	1.63	1.63
Br	–3.35	1.61	1.01	–6.03	2.26	1.97
Bl3	–3.53	1.65	1.16	–6.04	1.82	1.34
Bl4	–3.74	1.67	1.21	–6.79	1.795	1.31

**Table 2 materials-18-00631-t002:** Bonding energies $E_{bond}^{r}$ (eV), bonding distances $R_{bond}^{r}$ (Å) and distances to the surfaces $R_{surf}^{r}$ (Å) of H and O adatoms for various active sites of Pt(111), Pt(100), Pt(110) and Pt(211) surfaces.

	H			O		
Pt(111)	$E_{bond}^{r}$	$R_{bond}^{r}$	$R_{surf}^{r}$	$E_{bond}^{r}$	$R_{bond}^{r}$	$R_{surf}^{r}$
T	–3.67	1.55	1.55	–4.30	1.83	1.83
B	–3.51	1.77	1.18	–4.76	2.00	1.515
FCC	–3.48	1.88	1.08	–4.87	2.09	1.40
HCP	–3.44	1.88	1.07	–4.66	2.11	1.46
Pt(100)	$E_{bond}^{r}$	$R_{bond}^{r}$	$R_{surf}^{r}$	$E_{bond}^{r}$	$R_{bond}^{r}$	$R_{surf}^{r}$
T	–3.49	1.55	1.55	–4.52	1.80	1.80
B	–3.59	1.74	1.13	–5.57	1.94	1.405
H	–3.26	2.00	0.74	–4.95	2.185	1.08
Pt(110)	$E_{bond}^{r}$	$R_{bond}^{r}$	$R_{surf}^{r}$	$E_{bond}^{r}$	$R_{bond}^{r}$	$R_{surf}^{r}$
T1	–4.08	1.56	1.56	–5.56	1.79	1.79
Bl1	–4.26	1.75	1.13	–6.68	1.91	1.31
H1	–4.24	2.53	1.99	–5.79	2.73	2.22
H2	–4.22	2.06	1.51	–6.48	2.27	1.77
Bh1	–4.16	2.19	1.84	–5.80	2.40	2.07
Bh2	–4.17	2.18	1.85	–5.76	2.40	2.08
Bl2	–4.27	2.51	2.13	–6.33	2.91	2.59
T2	–3.85	1.71	1.71	–5.25	2.27	2.27
Pt(211)	$E_{bond}^{r}$	$R_{bond}^{r}$	$R_{surf}^{r}$	$E_{bond}^{r}$	$R_{bond}^{r}$	$R_{surf}^{r}$
Bh1	–3.80	1.75	0.99	–5.41	2.03	1.42
H1	–3.73	1.96	1.19	–5.10	2.10	1.40790
T1	–3.81	1.54	1.54	–4.89	1.82	1.82029
H2	–3.89	1.84	0.998	–6.49	2.02	1.32
Bh2	–4.08	1.72	0.94	–6.92	1.89	1.21
R	–3.92	2.205	1.03	–5.18	2.47	1.58
Bl1	–3.75	1.79	1.27	–5.18	1.97	1.48
Bl2	–3.83	1.80	1.30	–5.82	1.95	1.46
Hr	–3.75	2.14	1.56	–5.34	2.43	1.96
T2	–3.76	1.60	1.60	–5.31	1.90	1.90
H3	–3.76	1.83	0.98	–5.85	2.00	1.28
Bh3	–3.80	1.74	0.98	–6.21	1.96	1.34
H4	–3.80	1.96	1.26	–5.80	2.10	1.37
T3	–3.94	1.56	1.56	–5.51	1.77	1.77
Br	–3.75	2.20	1.79	–5.42	2.47	2.08
Bl3	–3.75	1.79	1.26	–5.18	1.97	1.55
Bl4	–3.83	1.77	1.29	–5.18	1.95	1.45

**Table 3 materials-18-00631-t003:** Bonding energies $E_{bond}^{r}$ (eV), bonding distances $R_{bond}^{r}$ (Å) and distances to the surfaces $R_{surf}^{r}$ (Å), of H and O adatoms for various active sites of Au(111), Au(100), Au(110) and Au(211) surfaces.

	H			O		
Au(111)	$E_{bond}^{r}$	$R_{bond}^{r}$	$R_{surf}^{r}$	$E_{bond}^{r}$	$R_{bond}^{r}$	$R_{surf}^{r}$
T	–2.12	1.58	1.58	–3.47	1.98	1.98
B	–2.55	1.78	1.28	–4.36	2.00	1.28
FCC	–2.57	1.88	1.12	–4.62	2.10	1.15
HCP	–2.63	1.89	1.01	–4.53	2.11	1.19
Au(100)	$E_{bond}^{r}$	$R_{bond}^{r}$	$R_{surf}^{r}$	$E_{bond}^{r}$	$R_{bond}^{r}$	$R_{surf}^{r}$
T	–3.00	1.596	1.596	–3.85	1.90	1.90
B	–3.21	1.77	1.15	–4.47	2.04	1.51
H	–3.28	2.01	0.65	–4.32	2.29	1.24
Au(110)	$E_{bond}^{r}$	$R_{bond}^{r}$	$R_{surf}^{r}$	$E_{bond}^{r}$	$R_{bond}^{r}$	$R_{surf}^{r}$
T1	–4.6	1.60	1.60	–7.19	1.88	1.88
Bl1	–4.78	1.79	1.20	–8.69	1.99	1.30
H1	–4.56	2.22	1.51	–7.98	2.39	1.81
H2	–4.71	2.02	1.37	–5.76	2.32	1.75
Bh1	–4.60	2.05	1.54	–7.97	2.26	1.65
Bh2	–4.59	2.00	1.57	–8.01	2.21	1.67
Bl2	–4.11	1.86	1.31	–5.29	2.43	2.03
T2	–3.89	1.63	1.63	–4.80	2.04	2.04
Au(211)	$E_{bond}^{r}$	$R_{bond}^{r}$	$R_{surf}^{r}$	$E_{bond}^{r}$	$R_{bond}^{r}$	$R_{surf}^{r}$
1Bh1	–3.02	1.81	1.11	–4.61	2.33	1.84
2H1	–3.11	2.18	1.46	–4.19	2.53	1.95
3T1	–3.10	1.60	1.60	–4.07	1.96	1.96
4H2	–3.37	1.90	1.04	–5.07	2.197	1.54
5Bh2	–3.53	1.74	0.997	–5.10	2.01	1.41
6R	–3.27	2.28	1.17	–5.21	2.59	1.70
7Bl1	–3.11	1.97	1.46	–4.18	2.32	1.81
8Bl2	–3.19	1.88	1.298	–4.53	2.16	1.65
9Hr	–3.15	2.27	1.75	–4.45	2.59	2.06
10T2	–3.16	1.66	1.66	–4.52	2.10	2.10
11H3	–3.31	1.936	1.07	–4.51	2.31	1.66
12Bh3	–3.31	1.77	1.04	–4.73	2.13	1.58
13H4	–3.19	2.06	1.27	–4.42	2.34	1.66
14T3	–3.22	1.59	1.59	–4.18	1.87	1.87
15Br	–3.15	2.30	1.95	–4.22	2.63	2.34
16Bl3	–3.12	1.98	1.45	–4.20	2.32	1.83
17Bl4	–3.18	1.94	1.35	–4.47	2.21	1.67

## Data Availability

The data presented in this study are available on request from the corresponding author.

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
