# Peer review of "Surface Structure Effects on H and O Adsorption on Gold, Nickel and Platinum Nanoparticles"

_materials, 2025, doi:10.3390/ma18030631_

Round 1
Reviewer 1 Report
Comments and Suggestions for Authors
Dear Editor,
The Authors here present their numerical investigations of the catalityc properties of a series of nano-particles and their binding energy for H and O. They identify the more favourable sites to use these as catalyst.
It is my opinion that the current manuscript should not be published as the Authors do fail in make their research clear.
1) For the sake of the analysis the nanoclusters are replaced by flat surfaces with different orientations, thus neglecting completely the local structure of the nano-particles. While this might be a starting point, it is not clear when this assumption should be valid for a nanocluster. Thus effectively, the research deals with the catalytic properties of surface of different coniage metals. In this respect, thus also the title might sound misleading.
2) The binding energy that are found seem overestimated. A binding energy of 6eV (H on Ni (110)) or 8 eV (O on Ni(110)) is possibly not correct since it would imply that Nichel exists only fullu oxydised.
3) If the aim of the work is to compare with available experiments, I would recommend collecting them in a representative figure where direct comparison would be possible.
4) Why the Authors use the d-band theory where there are more flexible tools (like the generalized coordination number) that might offer a better intepretation of their results?
5) The figures are of dismal quality. Figure 1d is simply a more chaotic restructuring of Figs. 1a, 1b and 1c. Figs 2,4 and 6 extends well over the page, have illegible labels and contain too much information to make any sense.
English is generally OK with some falls here and there that do not impede reading too much (but they should be cleaned).
That said, it is my informed opinion that the present article should not be published, that the title should be changed to better reflects the work done and that the presentation needs a thourough reworking.
Comments on the Quality of English LanguageThe manuscript is generally readable with some minor mistakes that can be amended in the eventual revision process.
Author Response
Many thanks for Your valuable comments on our manuscript.
- For the sake of the analysis the nanoclusters are replaced by flat surfaces with different orientations, thus neglecting completely the local structure of the nano-particles. While this might be a starting point, it is not clear when this assumption should be valid for a nanocluster. Thus effectively, the research deals with the catalytic properties of surface of different coniage metals. In this respect, thus also the title might sound misleading.
Answer: We have added relevant comments to the Introduction. The size of the real nanoparticles studied in the experiment was about 3–5 nm. Such nanoparticles consist of at least several hundred atoms. Such huge objects are beyond the capabilities of DFT modeling. On the other hand, it is quite possible to use small clusters as models of nanoparticles, although due to the limited size of the system, it may incorrectly reflect the features of the electronic structure desired for calculation. In this case, modeling nanoparticles using surfaces is the most acceptable approach, since the main emphasis here is on the qualitative comparison of changes in the electronic structures of model nanosystems in the experiment and simulation. In this work, a set of different surfaces simulates the different domains that make up the inhomogeneous surface of nanoparticles.
- The binding energy that are found seem overestimated. A binding energy of 6eV (H on Ni (110)) or 8 eV (O on Ni(110)) is possibly not correct since it would imply that Nichel exists only fullu oxydised.
Answer: DFT-modeling is known to be not an accurate tool for calculating binding energies or any other parameters, but it can be used to estimate trends and changes if the uniformity is maintained in models and the choice of simulation parameters. In this paper, emphasis was placed on finding relative changes in adsorption properties when varying adsorbates, surfaces, and elements. Simulation conditions are maintained constant in this case.
- If the aim of the work is to compare with available experiments, I would recommend collecting them in a representative figure where direct comparison would be possible.
Answer: The format of the data obtained using STM/STS experiments allowed comparison only at a qualitative level.
- Why the Authors use the d-band theory where there are more flexible tools (like the generalized coordination number) that might offer a better intepretation of their results?
Answer: We have added relevant comments to the Introduction. The results obtained from the STS measurements allow us to estimate the change in the local conductivity of nanoparticles during gas adsorption, i.e., the electronic structure. Similarly, when studying nanoparticle models using DFT-methods, the emphasis is on determining changes in the electronic structure of the models. Since the simulation is carried out on objects based on transition and noble metals, it is more convenient to use the d-band center model.
- The figures are of dismal quality. Figure 1d is simply a more chaotic restructuring of Figs. 1a, 1b and 1c. Figs 2,4 and 6 extends well over the page, have illegible labels and contain too much information to make any sense.
Answer: We have taken your wishes into account and have changed the Figs accordingly.
- The title should be changed to better reflects the work done.
Answer: We don’t agree with this comment and believe that the current title sufficiently reflects the essence of the work.

Reviewer 2 Report
Comments and Suggestions for Authors
Authors prepare to explain how the electronic struchture affects the adsorption properties of the nanoparticles by studing the flat and kinked surfaces of nickel, platinum and gold. The manuscript is worth to be published on Materials (ISSN 1996-1944) after revising some comments as following:
1. Could the authors explain why chose nickel, platinum and gold to study?
2. Please consider extending the introduction and add some related citations.
3. Please support some citations for formula 1,2,3,4 in line81 and line 95,97
4. Please label the meaning of the shapes in figure 1, and clarify figure2 which cannot read the labels.
Author Response
Many thanks for Your valuable comments on our manuscript.
- Could the authors explain why chose nickel, platinum and gold to study?
Answer: We have added relevant explanations to the Introduction. The choice of nickel, platinum and gold is due to two reasons. Firstly, these metals are widely used as a basis for model functional nanomaterials and were studied by our team using STM/STS methods. Secondly, the selected series of metals allows us to observe the relative change in adsorption properties when moving along the periodic table of elements from top to bottom from nickel to platinum and from left to right from platinum to gold.
- Please consider extending the introduction and add some related citations.
Answer: We agree with this idea and we have extended the Introduction accordingly.
- Please support some citations for formula 1,2,3,4 in line81 and line 95,97
Answer: We have added relevant references to the formula.
- Please label the meaning of the shapes in figure 1, and clarify figure2 which cannot read the labels.
Answer: We have taken Your wishes into account and have changed the Figs accordingly.

Reviewer 3 Report
Comments and Suggestions for Authors
Manuscript presented me to review process show a study about surface structure effects on H and O adsorption on gold, nickel and platinum nanoparticles. For this purpose the authors used a quantum-chemical tools. In my opinion the manuscript can be accepted in Materials, but first I recommend the improved it. Several point to consider:
1. In my opinion, the introduction section should include more information about: (I) the selection of (111), (100), (110), (211) surfaces and their characterization; (II) an explanation of why the authors only analyze the elements H and O.
2. Please consider adding a figure include an adsorption positions.
3. Please consider preparing a supporting information with data about bond length, vertical distance from the atom to the surface, and stretching frequency, and based on this show the most stable adsorption site.
Author Response
Many thanks for Your valuable comments on our manuscript.
- In my opinion, the introduction section should include more information about: (I) the selection of (111), (100), (110), (211) surfaces and their characterization; (II) an explanation of why the authors only analyze the elements H and O.
Answer: We have added relevant explanations to the Introduction. The choice of nickel, platinum and gold is due to two reasons. Firstly, these metals are widely used as a basis for model functional nanomaterials and were studied by our team using STM/STS methods. Secondly, the selected series of metals allows us to observe the relative change in adsorption properties when moving along the periodic table of elements from top to bottom from nickel to platinum and from left to right from platinum to gold. The set of surfaces was chosen based on the variation of the surface curvature (flat / kinked) and the possibility of modeling the cell within the DFT-framework. The supercells of high-index surfaces, for example (321), contain over a hundred atoms, so two surfaces containing various features were chosen as high-index surfaces: (110) with a break, that is, alternating chains of protruding and depressed atoms, and (211) with a step, that is, an edge formed by low-coordinated atoms. Smooth surfaces (111) and (100) were used to model homogeneous sets of active sites with different packing: dense (111) and more sparse (100). When choosing the adsorbate, we also took into account the results of experiments where hydrogen and oxygen acted as a simple model adsorbate. In the simulations, two different adsorbates also made it possible to compare the adsorption properties of surfaces with adatoms of different types.
- Please consider adding a figure include an adsorption positions.
Answer: We have taken your wishes into account and have changed the Figs accordingly.
- Please consider preparing a supporting information with data about bond length, vertical distance from the atom to the surface, and stretching frequency, and based on this show the most stable adsorption site.
Answer: We have added relevant data to the Tables 1–3
